# Efficient Data Learning for Open Information Extraction with Pre-trained Language Models

**Zhiyuan Fan**[1,2] and **Shizhu He**[1,3*]

[1]The Laboratory of Cognition and Decision Intelligence for Complex Systems,
Institute of Automation, Chinese Academy of Sciences, Beijing, China
[2]Tianjin University, Tianjin, China
[3]School of Artificial Intelligence, University of Chinese Academy of Sciences, Beijing, China
`zhiyuan.fan2002@gmail.com, shizhu.he@nlpr.ia.ac.cn`

## Abstract

Open Information Extraction (OpenIE) is a fundamental yet challenging task in Natural Language Processing, which involves extracting all triples (subject, predicate, object) from a given sentence. While labeling-based methods have their merits, generation-based techniques offer unique advantages, such as the ability to generate tokens not present in the original sentence. However, these generation-based methods often require a significant amount of training data to learn the task form of OpenIE and substantial training time to overcome slow model convergence due to the order penalty. In this paper, we introduce a novel framework, OK-IE, that ingeniously transforms the task form of OpenIE into the pre-training task form of the T5 model, thereby reducing the need for extensive training data. Furthermore, we introduce an innovative concept of **Anchor** to control the sequence of model outputs, effectively eliminating the impact of order penalty on model convergence and significantly reducing training time. Experimental results indicate that, compared to previous SOTA methods, OK-IE requires only 1/100 of the training data (900 instances) and 1/120 of the training time (3 minutes) to achieve comparable results.

## 1 Introduction

Open information extraction (Banko et al., 2007) is the task of transforming unstructured text into semi-structured text. Given an input sentence, an open information extraction system can output a set of corresponding sentence with elements in a pre-defined structure (Etzioni et al., 2011). These output can be used as an additional source of information to augment other tasks, such as building semi-automated knowledge graph construction systems (Mausam, 2016), Question Answering (Khot et al., 2017), Trigger Detection (Dukić et al., 2023) and so on.

Recently, neural-based OpenIE methods can be broadly divided into two categories: labeling-based methods (Kolluru et al., 2020a; Vasilkovsky et al., 2022; Kotnis et al., 2022) and generation-based methods (Kolluru et al., 2020b, 2022). Labeling-based methods are known for their fast inference speed but have a limitation in that they cannot label tokens that are not present in the input sentence. On the other hand, generation-based methods possess the capacity to produce supplementary tokens in order to fulfill syntactic and comprehensive semantic requirements, albeit necessitating extensive training data and time investment to learn the task format of OpenIE and converge towards desirable outcomes.

GEN2OIE (Kolluru et al., 2022) has set a robust baseline for generation-based methods by modeling the OpenIE task as a seq2seq problem using two T5 (Raffel et al., 2022) models. The first T5 model is employed to extract all predicates from the sentence, while the second T5 model takes a sequence concatenated from the sentence and an individual predicate as input, outputting the corresponding unique triple. While the GENO2IE sets a strong baseline, there are still areas for improvement. We address these with our proposed OK-IE framework, an **O**rder Agnostic **K**nowledge Embedded **I**nformation **E**xtraction framework. Considering the simplicity of predicate extraction from sentences, we retain the first stage of GEN2OIE and focus on optimizing the second stage, which is responsible for generating the complete triple.

Specifically, unlike GEN2OIE, OK-IE cleverly transforms OpenIE into T5's pre-training task, span corruption. This allows the model to learn the task with less data, eliminating the need for additional training to adapt the language knowledge from span corruption pre-training to the seq2seq. In addition, the seq2seq presumes that only one sequence order is correct. This order information is not explicitly provided to the model, which must learn it on

---
*Corresponding author.

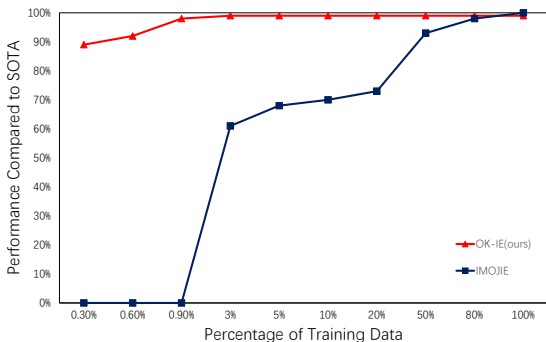

Figure 1: The performance of both methods improves as the volume of data increases. Compared to our OK-IE, IMOJIE needs to use about 83(50% vs 0.6%) to 89(80% vs 0.9%) times more data to achieve approximately the same results.

| OpenIE System | F1 | F1% |
|---|---|---|
| **IMOJIE** | 53.5 | 99.07 |
| **OpenIE6** | 52.7 | 97.6 |
| **Multi²OIE** | 52.5 | 97.2 |
| **IGL-OIE** | 52.4 | 97.03 |
| **CIGL-OIE** | **54.0** | **100** |
| **DetIE** | 52.1 | 96.4 |
| **GEN2OIE** | 51.1 | 94.6 |
| **OK-IE (Ours)** | 53.2 | 98.5 |

Table 1: Comparison of OpenIE systems with full data (100%)

its own. This leads to an additional order penalty, causing confusion for the model and requiring a long time to converge. OK-IE addresses this issue by introducing an **Anchor** method, explicitly instructing the model on the sequence order for generation. The comprehensive workflow of OK-IE are elaborated in the Appendix A.

The effectiveness of our proposed OK-IE framework is exemplified particularly in low-resource scenarios on the CaRB (Bhardwaj et al., 2019) benchmark, as illustrated in Figure 1. Summarizing, our key contributions are:

1) We introduce the OK-IE framework that harnesses the potential of PLMs for efficient OpenIE. It uses minimal data and computational resources, an important attribute in resource-limited scenarios, while achieving results comparable to full-data training.

2) We address the order challenge within the OK-IE framework, allowing for faster training convergence by eliminating the additional order penalty.

3) Our OK-IE framework not only achieves SOTA performance in a low-resource scenario but also presents a new perspective for OpenIE research, potentially inspiring more efficient and effective approaches in this area.

## 2   Related Work

Predominant solutions to the OpenIE task fall into two categories: labeling-based and generation-based methods. This paper aims to address the excessive requirements of training data and time inherent in generation-based methods, and thus we will primarily focus on discussing generation-based techniques. Labeling-based methods, while important, will be briefly overviewed for the sake of context.

**labeling-based** methods aim to tag each token in a sentence with one of four labels: subject, predicate, object, or none. Representative work includes OpenIE6 (Kolluru et al., 2020a), which initially marks the first triple in a sentence, then merges this marked triple with the original sentence to form a new input for identifying the second triple, iterating in this manner until termination. DetIE (Vasilkovsky et al., 2022), on the other hand, borrows ideas from object detection in computer vision to identify all triples in a sentence in one go. Nonetheless, these techniques encounter a shared challenge: they lack the ability to label tokens absent in the original sentence. This shortcoming hampers their ability to invariably yield triples that satisfy grammatical coherency and semantic completeness.

**Generation-based** approaches present an alternative perspective in OpenIE, treating both the input sentences and output triples as sequences. For example, IMOJIE (Kolluru et al., 2020b), leveraging an LSTM model, sequentially generates triples. It takes an input sentence and produces the first triple, then concatenates this generated triple with the original sentence to form a new sequence. The process continues iteratively until the system reaches an end token. GEN2OIE (Kolluru et al., 2022), a powerful baseline in our study, employs a dual T5 model setup. The first T5 model is responsible for generating all predicates from a given sentence. Subsequently, the second T5 model accepts the concatenation of the sentence and an individual predicate as input, producing the corresponding triple for that predicate. However, these techniques generally require extensive training data and time to yield satisfying results.

Despite GEN2OIE's utilization of the pre-trained language model T5 as its backbone, a considerable discrepancy exists between the seq2seq task and T5's pre-training task, span corruption. The model still necessitates an ample amount of data to learn this new task format. Concurrently, in the context of triples, both (subject, predicate, object) and (predicate, object, subject) orders are valid. However, due to the auto-regressive characteristic inherent in sequence generation, training presumes only one correct generation order. Moreover, this assumed order is implicit and not directly conveyed to the model. Consequently, the model might generate a correctly composed triple in an order inconsistent with the provided ground truth, resulting in an additional order penalty. This discrepancy can lead to model confusion, slow convergence, and thus, requires substantial training time.

## 3 Method

In addressing the aforementioned issues, our proposed framework, OK-IE, pivots around two core aspects: the transformation of task format and the control over generation order. To better control the generation order, we introduce **Anchors** and merge them with sentinels.

### 3.1 Task Format Transformation

Seq2Seq are designed to model relationships between two sequences. However, the T5 model is trained with a span-mask denoising objective. For instance, when the input is 'Thank you <id_0> me to your party <id_1> week,' the expected output is '<id_0> for inviting <id_1> last.' Here, <id_i> are referred to as sentinels, which serve to locate the positions corrupted in the original sentence. Notably, we discerned an opportunity to ingeniously transform the OpenIE task to match the span corruption objective. More specifically, the input sentence and output triples are concatenated into one sentence, treating each subject, predicate, and object in the triple as a corrupted span. For instance, under the Seq2Seq task format, the input and output are: 'Elon Musk, who is the CEO of Tesla, also founded SpaceX', and '(Elon Musk; is the CEO of; Tesla) (Elon Musk; founded; SpaceX)', respectively. After the task format transformation, the input and output become: 'Elon Musk, who is the CEO of Tesla, also founded SpaceX. With

predicate founded, <id_0> <id_1> <id_2> . With predicate is the CEO of, <id_3> <id_4> <id_5> ', and '<id_0> Elon Musk <id_2> founded <id_2> SpaceX <id_3> Elon Musk <id_4> is the CEO of <id_5> Tesla', respectively. The efficacy of this seemingly simple yet innovative method is demonstrated in subsequent experiments.

### 3.2 Controlling Generation Order

As evidenced by the aforementioned examples, it is clear that sentinels such as '<id_i>' only denote the locations of corrupted spans within sentences. Due to the reusable nature of sentinels, it's difficult to assign specific meanings to each sentinel. Hence, in the example given, we still implicitly assume a particular order. To mitigate the impact of order penalty, we introduce the concept of Anchors which temporarily assign meanings to each sentinel to help explicitly control the generation order.

An anchor refers to tokens surrounding the sentinel, intended to express the sentinel's temporary meaning. More specifically, if we want the sentinel '<id_0>' to generate only subjects and nothing else, we can add anchors representing the need to generate a subject on either side of the sentinel. Each anchor consists of two parts: an embedding from a word table representing a clear meaning, and a tunable embedding(Liu et al., 2021); the aforementioned embeddings are concatenated to form a cohesive semantic unit. For simplicity, we denote the anchors for subjects, predicates, and objects as S, P, and O, respectively. It's worth noting that the same anchor is used for each subject in all triples of a sentence, ensuring that the tunable embeddings can learn more general semantic representations.

By integrating anchors and sentinels, we can explicitly control the model's generation order. For instance, in the original example, <id_0> <id_1> <id_2> implies an order of subject, predicate, object. If we want to make this order explicit, we can use 'S<id_0>S P<id_1>P O<id_2>O'. If we prefer the model to generate in the order of predicate, object, subject, it would be 'P<id_0>P O<id_1>O S<id_2>S'. Note that, in practical applications, we can select an appropriate order through practice, or we can control the model to generate all orders and then use a majority vote strategy to determine the final output. In Figure 2, we illustrate how anchors are employed to control token generation at each sentinel position.

## 4 Experiments

### 4.1 Experimental Setup

| OpenIE System | F1 | F1% |
|---|---|---|
| IMOJIE | - | - |
| OpenIE6 | 42.9 | 79.4 |
| OK-IE | **52.9** | **97.9** |

Table 2: Comparison of OpenIE systems with 0.9% data

| OpenIE System | F1 | F1% |
|---|---|---|
| IMOJIE | 36.9 | 68.5 |
| OpenIE6 | 43.1 | 79.8 |
| OK-IE | **53.1** | **98.3** |

Table 3: Comparison of OpenIE systems with 5% data

| OpenIE System | F1 | F1% |
|---|---|---|
| IMOJIE | 39.5 | 73.2 |
| OpenIE6 | 45.3 | 83.8 |
| OK-IE | **53.3** | **98.7** |

Table 4: Comparison of OpenIE systems with 20% data

To establish an objective and fair comparison with other models, we opted for the benchmark dataset, CaRB (Bhardwaj et al., 2019), used in previous studies, applying the same evaluation metric, F1 score. For consistency, we utilized the previously established OpenIE system training dataset, IMOJIE (Kolluru et al., 2020b). Owing to space limitations, further experimental details are provided in Appendix B.1.

### 4.2 Results and Discussion

To visually demonstrate the performance of OK-IE, we compared it with several recent OpenIE systems under full training data scenarios, as shown in Table **??**. To facilitate a straightforward comparison between different methods, we introduce the F1% metric. This represents the F1 score of a given method within a specific data scenario, expressed as a percentage of the F1 score of the SOTA method when trained with full data. This metric simplifies performance comparisons in low-resource scenarios. As can be seen, the F1 scores of all methods are comparable. A significant advantage of OK-IE, however, is its ability to substantially reduce the required training resources. To investigate the performance of OK-IE in low-resource situations, we set three data sizes relative to the full data set:

| Approach | F1* | F1 |
|---|---|---|
| Baseline | 29.9 | 31.2 |
| + Convert Form | 45.3(**+15.4**) | 49.8(**+18.6**) |
| + Anchor | 46.0(+0.7) | 50.5(+0.7) |
| + Order Control | **52.1**(+6.1) | **52.9**(+2.4) |

Table 5: Results of ablation experiments on various components of framework OK-IE.

0.9%, 5%, and 20%. We selected the most representative models from both labeling-based and generation-based methods, namely OpenIE6 and IMOJIE, for comparison. From Tables 2, 3, and 4, it is evident that our OK-IE markedly outperforms the other two methods across all three data sizes. It's worth noting that due to the minimal data availability at 0.9%, IMOJIE is incapable of generating effective triples, hence its result is denoted by -.

### 4.3 Ablation Studies on System Components

In order to analyze the impact of individual components of OK-IE on the results with a finer granularity, we carried out an ablation study based on the baseline GEN2OIE in a scenario with only 900 data items. Simultaneously, to validate the influence of order control on model convergence speed, we defined a new metric, F1*, which refers to the F1 score obtained after just one epoch of training. As can be seen from the Table 5, a noticeable enhancement in the optimal F1 is achieved following the transformation of the task form, which indeed confirms a decrease in training data requirement. Upon introducing order control, there is a swift increase in F1*, and the result after only one epoch of training is very close to the optimal F1 score. This confirms that model convergence speed has indeed been accelerated, thus reducing the training time. For details on the analysis of training data and training time, please refer to the Appendix B.2.

## 5 Conclusion

In this paper, we have presented OK-IE, an efficient framework for Open Information Extraction. By effectively leveraging pre-trained language models and addressing the challenges of generation order, OK-IE overcomes the computational inefficiencies of previous systems. Our empirical evaluations demonstrate that OK-IE not only achieves comparable results to existing systems, but does so with significantly fewer resources. This underlines its capability in resource-constrained environments.

Our research stands as a testament to the possibility of accomplishing efficient information extraction with reduced data and minimal training time, thereby presenting new pathways for future research within the OpenIE landscape.

## Limitations

Following an examination of the CARB test set and the output generated by existing OpenIE systems, we found that these systems struggle to effectively manage cases that involve extensive sentences with multiple triples.

While OK-IE has the ability to enhance computational efficiency and the optimization of data resources, it does not sufficiently address the aforementioned issues. The generation-based strategy is capable of creating long, meticulously refined triple sequences, but there is still the possibility that some triples may be omitted. Similarly, the conventional labeling-based approach adopted in OpenIE6 sets an upper threshold on the number of triples that require labeling, hence capping its performance. Accurately generating all triples in extended sentences represents a promising avenue for future research.

A potential pitfall for OK-IE, given its modest data requirements, is the potential for bias in scenarios characterized by substantial differences in data characteristics. However, this issue can be tackled by the framework itself, as OK-IE exhibits the ability to rapidly adapt to new scenarios and ameliorate these biases with a minimal amount of data.

It should be underscored that our current method has been specifically evaluated within the confines of English OpenIE. This presents a limitation, and we intend to explore and validate the efficacy of our approach across other languages in future work.

## Ethics Statement

Our proposed OK-IE system has the potential to contribute to the construction of comprehensive knowledge databases. Furthermore, given the stringent context window size constraints associated with applications like ChatGPT, the use of triples extracted by OK-IE could effectively alleviate text length in input. Significantly, one of the notable advantages of our OK-IE system is its ability to achieve commendable performance with minimal resource requirements, enhancing its accessibility and practicality for a wide range of users and applications.

However, it should be noted that there are circumstances under which OK-IE may output erroneous triples, leading to serious inaccuracies in the knowledge databases or incorrect triple inputs to applications like ChatGPT. These errors could, in turn, result in significant misconceptions, as flawed knowledge produced upstream could greatly damage downstream applications, potentially leading to misdirection in human activities.

Therefore, we strongly urge users of our system to implement rigorous human oversight and checks on extracted knowledge, while fully evaluating the potential harm that could be caused by inaccuracies. Our primary ethical responsibility lies in minimizing any potential for misuse of the system or misinterpretation of the output, to ensure that it serves the purpose of advancing knowledge rather than misleading users.

## Acknowledgements

This work was supported by the National Key R&D Program of China (2022ZD0160503) and the National Natural Science Foundation of China (No.62376270, No.61831022). This work was supported by the Strategic Priority Research Program of Chinese Academy of Sciences (No.XDA27020100), Youth Innovation Promotion Association CAS and OPPO Research Fund.

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

## A Overview of OK-IE

### A.1 Workflow

The extraction process employed by OK-IE is both intuitive and flexible. Initially, an input sentence is introduced, and a T5 model is employed to extract all predicates within the sentence. Subsequently, as illustrated in Section 3, the sentence and its extracted predicates are combined, with expected triple positions replaced by sentinel <id_i> placeholders. This combination is then fed into a second T5 model, which generates the appropriate span for each sentinel-marked position, filling in the respective subject, predicate, or object. It is important to note that the order in which OK-IE extracts the final results is entirely dependent on the training set.

To manage the generation order, we rely on anchors. For example, if the extraction is intended to proceed in a predicate, object, subject order, the combination of anchor and sentinel would result in P<id_0>P O<id_1>O S<id_2>S.

Two key aspects should be noted. Firstly, the optimal extraction order can be determined through

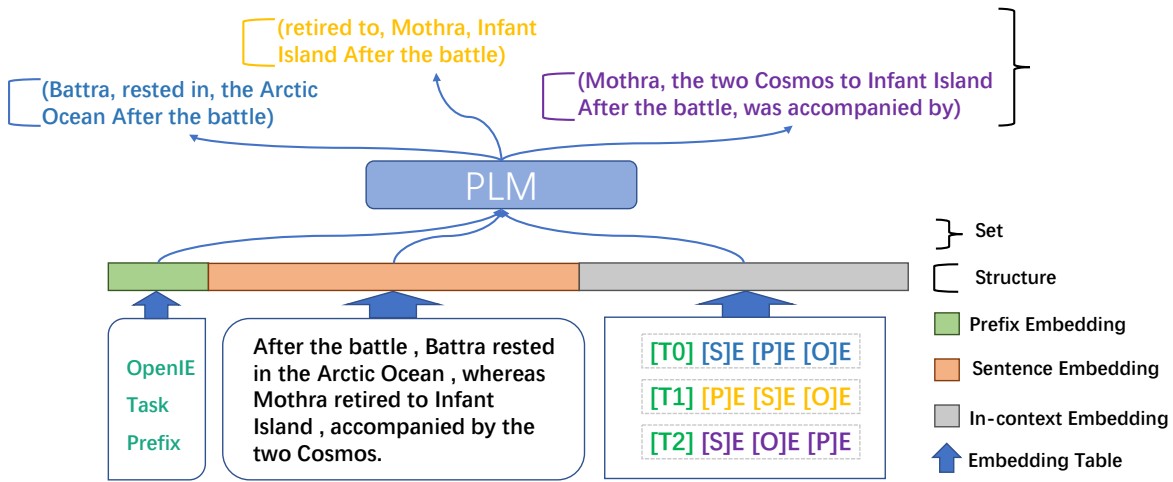

Figure 2: Workflow of OK-IE. In the figure, 'E' denotes the sentinel, while [S], [P], and [O] represent anchors for subject, predicate, and object respectively. Blue, yellow, and purple lines illustrate three distinct generation sequences: SPO, PSO, and SOP. By integrating anchors and sentinels, control over token generation at each position is achieved, thereby governing the generation order of individual triples.

trial and error in practical scenarios, or by generating results under all extraction orders and utilizing a majority vote strategy to determine the final extraction results. Secondly, for a given sentence, the order of the triples to be extracted can vary. If we wish to extract the triple for the first predicate in a subject, predicate, object order, and the triple for the second predicate in a predicate, object, subject order, it can conveniently be incorporated in the same generation process: ... S<id_0>S P<id_1>P O<id_2>O ... P<id_3>P O<id_4>O S<id_5>S. This flexible generation method opens up a myriad of possibilities.

### A.2 Difference

Several distinct differences exist between OK-IE and GEN2OIE:

Firstly, GEN2OIE approaches the OpenIE problem using a seq2seq task format, while OK-IE employs a span corruption task format. This difference in task format leads to a substantial variation in the amount of training data required during the training process.

Secondly, the order of the triples extracted by GEN2OIE is determined by the training data and can only follow one specific order, i.e., the order of the triples in the training data. Conversely, OK-IE, through the introduction of the **Anchor**, can flexibly determine the extraction order of the triples in accordance with the actual scenario, allowing for the coexistence of multiple extraction orders.

Next, GEN2OIE can only extract a single triple

corresponding to a predicate each time, whereas OK-IE can extract either one or all triples corresponding to the predicates, depending on how the anchor and sentinel are set. This contributes to the differences in model computation efficiency.

Lastly, to extract triples using a generation-based method, GEN2OIE introduces additional delimiters to denote the boundaries of the subject, predicate, and object. In contrast, OK-IE utilizes native sentinels, effectively sidestepping this unnecessary issue.

## B Experiments

### B.1 Experimental Setup Details

In this study, we employed the PyTorch 1.11 (Paszke et al., 2019) framework to conduct our experiments. Specifically, we utilized the base series of the T5 model from the Hugging Face's transformers (Wolf et al., 2020) library, without incorporating any additional parameters, except for word embeddings. The cross-entropy loss function and Adam (Kingma and Ba, 2017) optimizer were utilized for training. All models were trained for a consistent duration of 7 epochs.

With regards to hyperparameter selection, we investigated batch sizes of (2, 4, 8, 16, 32) and ultimately chose a batch size of 4 to optimize memory usage while having no significant impact on the results. For the learning rate, we experimented with values of (5e-5, 2e-5, 1e-5, 5e-6) and selected 5e-5 as the optimal value. All other parameters were kept at their default values and we did not use any

form of hyperparameter search algorithm. In order to mitigate the influence of random variance, we conducted three separate random samplings under each limited training data scenario and reported the mean values as final results.

As previously mentioned, OK-IE can specify any order for generating extraction results by designing anchors. During the training process, we utilized all possible orders of triples to train the model. As a side note, if GEN2OIE were to adopt this strategy, it would result in significant model confusion. In the evaluation phase, to avoid the influence of additional factors, we did not use the strategy of generating extraction results in all possible orders and then using a majority vote to obtain the final output. Instead, we explicitly set a unique extraction order, i.e., S<id_0>S P<id_1>P O<id_2>O.

### B.2 Training Data and Time Analysis

Evidently, conducting a direct analysis of the quantity of training data and training time required by two models to achieve identical performance levels is a complex undertaking. On one hand, the correlation between training data and time is non-negligible; in most cases, a reduction in data volume correlates with decreased training time. On the other hand, achieving parity in evaluation results between the two models can be challenging. Thus, the previously mentioned ratios of 1/100 for training data and 1/120 for training time were derived directly from comparing the resources required by OK-IE to achieve comparable results with GEN2OIE on the full dataset, but with only 900 instances. While we could theoretically start from 900 and gradually reduce the amount of training data to see how performance fluctuates, the potential findings from such an approach would likely be marginal.

In order to accurately quantify the model's demands for training data and time, we set up various scenarios with limited training data and selected some representative models to compare their standout results within these contexts. Regarding training time, in order to impartially offset the impact of training data volume and focus on the influence of methodological improvements, we kept the training data fixed. We gauged the model's convergence speed by evaluating the F1 score (F1*) after a single epoch of training. In other words, a larger F1* suggests quicker model convergence, indicating that the methodological improvements have effec-

| Model | Task Format | F1 |
|---|---|---|
| flan-t5-base | seq2seq | 0.0021 |
| flan-t5-large | seq2seq | 0.244 |
| flan-t5-xl | seq2seq | 0.2852 |
| t5-base | seq2seq | - |
| t5-large | seq2seq | - |
| t5-xl | seq2seq | 0.0016 |
| t5-base | span corruption | 0.0007 |
| t5-large | span corruption | 0.0019 |
| t5-xl | span corruption | 0.0023 |

Table 6: Zero Shot Performance

tively reduced the required training time for the model.

## C  Zero Shot Performance

We employed the GEN2OIE model to generate predicates in the first phase, following which we conducted zero-shot inference on t5 models across three size attributes: base, large, and XL, as well as on the flan-t5 model. The outcomes of these experiments are reported in Table 6. Due to the incapability of flan-t5 to generate in the form of span corruption, we only report the results under the seq2seq task paradigm.