# OpenReview forum: "Efficient Data Learning for Open Information Extraction with Pre-trained Language Models"
_EMNLP/2023/Conference — EMNLP 2023 Findings_

### Official Review · Reviewer_DMKQ · 2023-08-05

**Typos Grammar Style And Presentation Improvements:** pg 4 col 2 line 295
**Soundness:** 4

**Excitement:**

4: Strong: This paper deepens the understanding of some phenomenon or lowers the barriers to an existing research direction.

**Paper Topic And Main Contributions:**

The present paper proposes an encoding of OIE as a data corruption task and employs a T5 model to perform the triple generation process of the GEN2OIE model by Kolluru et al. (2022) in a few-shot scenario. While the idea of generation-based OIE is not innovative, and their architecture is based on Kolluru et al.'s model, the paper relies on a very ingenious encoding of the task in a way that reduces the computational complexity of training the T5 triple generation model by requiring only a fraction of the training examples. The authors conduct an empirical validation of their approach and ablation studies to show how each aspect contributes to the performance, Particularly, they evaluate the impact of the encoding of the task as span prediction by investigating how the model converges to a result after one single training epoch.


**Reasons To Accept:**

I think the work is fairly clear, and the results are interesting. It is not groundbreaking in its approach, but it is not trying to be, and the authors are very clear on the objectives of their work.
The work relies on a simple but effective idea of encoding triple generation as span corruption and leveraging T5 to learn to predict accurately for the English language. Their empirical validation is consistent and their analysis are insightful on how each part contributes to the performance


**Reasons To Reject:**

 The major limitation, in my opinion, is its reliance on the English language. English is relatively poor in terms of syntax, and for a task that is so dependent on syntax, such as OIE, the performance of the method may stem from the language characteristics. Notice that Kolluru et al. have investigated GEN2OIE in a multilingual setting, which gives us a better understanding of the performance and limitations of the work. Also, since the authors are limited to the English language, in my opinion, they should very explicitly state that, as their results may simply not generalise to OIE as a whole but to OIE in English.

**Reproducibility:**

5: Could easily reproduce the results.

**Reviewer Confidence:**

4: Quite sure. I tried to check the important points carefully. It's unlikely, though conceivable, that I missed something that should affect my ratings.

---

> ### Author Rebuttal · Authors · 2023-08-28
>
> **We sincerely appreciate the time and effort you dedicated to reviewing our work. Below, we respond to each of your comments and questions.**
>
> ---
>
> ### Question 1:
> The major limitation, in my opinion, is its reliance on the English language. English is relatively poor in terms of syntax, and for a task that is so dependent on syntax, such as OIE, the performance of the method may stem from the language characteristics. Notice that Kolluru et al. have investigated GEN2OIE in a multilingual setting, which gives us a better understanding of the performance and limitations of the work. Also, since the authors are limited to the English language, in my opinion, they should very explicitly state that, as their results may simply not generalise to OIE as a whole but to OIE in English.
>
> ### Answer 1:
> Thank you for your discerning observation. In the limitation section of our paper, we primarily discuss the challenge of extracting from complex sentences containing multiple triples. We concur with your perspective and will amend our limitation section to explicitly state that our method has been specifically evaluated in the context of English OpenIE. Explorations and adaptations of our proposed approach for other languages are earmarked as a crucial direction for our future endeavors.
>
> Furthermore, in the revised version, post line L348 in the limitation section, we have added: "It should be underscored that our current method has been specifically evaluated within the confines of English OpenIE. This presents a limitation, and we intend to explore and validate the efficacy of our approach across other languages in future work."
>
> ---
>
> ### Question 2:
> pg 4 col 2 line 295: table 5 -> Table 5
>
> ### Answer 2:
> Thank you for your meticulous attention to detail. You are right, and we acknowledge the oversight, it was a typographical error regarding capitalization during typing. We have corrected this error in the revised version.
>
> Besides, we will check the entire paper for any potential typo errors.

---

### Official Review · Reviewer_8h8N · 2023-08-11

**Soundness:** 4

**Excitement:**

4: Strong: This paper deepens the understanding of some phenomenon or lowers the barriers to an existing research direction.

**Paper Topic And Main Contributions:**

This paper presents a study on low resource open information extraction and introduces a novel approach to tackle the issue of data inefficiency. The authors propose a shift in the task paradigm of the second stage of GENO2IE, opting for a span corruption paradigm instead. This paradigm change helps overcome the data inefficiency caused by the pretraining and fine-tuning gap. Additionally, the authors suggest the use of anchors to guide the generation order, addressing a critical challenge in Open IE.

**Questions For The Authors:**

1. Can the paradigm shift work in a more general form (i.e. extract the triplets end2end without the step1 of GENO2IE) ？

**Reasons To Accept:**

1. The paper introduces a paradigm shift that is technically sound and demonstrates good performance in low resource scenarios.
2. The experimental results provide evidence that the proposed anchor guidance and order control techniques are effective in enhancing performance and alleviating the order problem.

Overall, the contributions of this paper are suitable enough to present as a short paper.

**Reasons To Reject:**

1. Although the idea of anchors can be roughly understand through careful reading. Some implementation details about the anchor guidance are missing. The description between L226 and L229 are not enough for presenting the anchor to all readers, especially the p-tuning process of anchors is missing.

**Reproducibility:**

4: Could mostly reproduce the results, but there may be some variation because of sample variance or minor variations in their interpretation of the protocol or method.

**Reviewer Confidence:**

3: Pretty sure, but there's a chance I missed something. Although I have a good feel for this area in general, I did not carefully check the paper's details, e.g., the math, experimental design, or novelty.

**Typos Grammar Style And Presentation Improvements:**

If the idea of anchors can be presented in the form of a small figure (even in the Appendix), it would be better.

---

> ### Author Rebuttal · Authors · 2023-08-28
>
> **We sincerely appreciate the time and effort you dedicated to reviewing our work. Below, we respond to each of your comments and questions.**
>
> ---
>
> ### Question 1:
> Can the paradigm shift work in a more general form (i.e. extract the triplets end2end without the step1 of GENO2IE) ?
>
> ### Answer 1:
> Certainly, your question is astute and insightful. Indeed, the paradigm can work in a more general form, enabling end2end extraction of triples without the preliminary step of GENO2IE.
>
> However, from our prior experiments, we observed that direct end2end extraction utilizing the paradigm shift didn't yield as significant improvements as when the shift was employed in the pipeline approach. To provide a specific instance, when leveraging the paradigm shift within the pipeline under a low-resource setting, we achieved an F1 score of 52.9. In contrast, the end2end approach with the paradigm shift resulted in an F1 of 39.7. It's crucial to note that this increment is still considerable, especially given that without the paradigm shift, the end2end approach performs rather poorly under low resource conditions.
>
> Such findings are undeniably intriguing. One reason we didn't discuss this aspect in the current paper, along with the associated data, is due to the evident performance disparity. In the pipeline extraction method—where we first extract the predicate and then the entire triple—the predicate extraction phase is relatively simpler. Therefore, when viewed holistically, we opted to forgo the convenience of the end2end process in favor of more significant performance gains in the pipeline approach. It's a conscious tradeoff we decided to make based on empirical evidence.
>
> ---
>
>
> ### Question 2:
> Although the idea of anchors can be roughly understand through careful reading. Some implementation details about the anchor guidance are missing. The description between L226 and L229 are not enough for presenting the anchor to all readers, especially the p-tuning process of anchors is missing.
>
> ### Answer 2:
>
> To elucidate the concepts of the Anchor and the p-tuning process more thoroughly, we offer the following elaboration:
>
> - **Anchor**:
>   As delineated in lines `L226-L229` of the paper,
>   >"Each anchor consists of two parts: an embedding from a word table representing a clear meaning, and a tunable embedding (Liu et al., 2021)."
>
>   In this context, the former refers to a fixed vector, obtained from the word embedding layer of a language model, which represents a distinct semantic meaning. This vector is derived from token ids that are acquired through tokenization (specifically within the transformers library as `input_ids`). The latter, on the other hand, refers to a soft prompt embedding. This is an initially set vector that remains tunable, encapsulating a latent semantic meaning.
> - **P-tuning process**:
>   Within the implementation, the aforementioned embeddings are concatenated to form a cohesive semantic unit. This unit explicitly defines the type (subject, predicate, object) to be generated at each sentinel's position. Specifically, if we intend to produce a 'subject' at the `<extra_id_0>` (the sentinel symbol in the transformers library for T5) position , the anchor comprises the vector derived from the word embedding layer for the token “subject”, combined with a tunable vector. This anchor is then placed on both sides of the sentinel's embedding to constrain the contextual surroundings of this position to 'subject'. The result is an `input_embeds` that is fed into the model.
>
> We believe that this detailed explanation, complemented by specifics from the implementation, will make the concepts under discussion more comprehensible. We have augmented this section in the appendix of the revised version.
>
> ---
>
>
> ### Question 3:
>
> If the idea of anchors can be presented in the form of a small figure (even in the Appendix), it would be better.
>
> ### Answer 3:
>
> Thank you for your valuable input. In response to your suggestion, we have completed a figure that elucidates the concept of anchors. This illustration, accompanied by the details mentioned in A2, will be included in the Appendix for enhanced clarity.

---

### Official Review · Reviewer_iPLc · 2023-08-11

**Soundness:** 1

**Excitement:**

1: Poor: I cannot identify the contributions of this paper, or I believe the claims are not sufficiently backed up by evidence. I would fight to have it rejected.

**Paper Topic And Main Contributions:**

They present a method called OK-IE that is supposed to transform Open Information Extraction task into a pretraining task form for T5.

**Reasons To Accept:**

If the results are correct, it could be very useful.

**Reasons To Reject:**

The paper is too short and not well explained. I cannot understand the key component of the proposed idea, neither the working flow.

The experimental setup is also weak. There is a single dataset. The tables could be just integrated in Figure 1. The ablation experiment is also unclear for the reader.

**Reproducibility:**

2: Would be hard pressed to reproduce the results. The contribution depends on data that are simply not available outside the author's institution or consortium; not enough details are provided.

**Reviewer Confidence:**

2: Willing to defend my evaluation, but it is fairly likely that I missed some details, didn't understand some central points, or can't be sure about the novelty of the work.

---

> ### Author Rebuttal · Authors · 2023-08-28
>
> **We sincerely appreciate the time and effort you dedicated to reviewing our work. Below, we respond to each of your comments and questions.**
>
> ### Question 1:
> There is a single dataset.
>
> ### Answer 1:
> Importantly, it should be emphasized that representative works in the Open Information Extraction domain, such as **OpenIE6 (presented at EMNLP)**, **IMOJIE (at ACL)**, and **GEN2OIE (at ACL)**, have all solely relied on the CaRB dataset for their evaluations. This singular choice was underscored in the OpenIE6 paper, **Section 6, "Experimental Setup"**, where it's mentioned,
> >"We evaluate all systems against CaRB’s reference extractions, as they have higher coverage and quality compared to other datasets."
>
> Indeed, as detailed in Section 4.1 "Experimental Setup" of our paper,`lines L251-L255`, we stated,
> >"To establish an objective and fair comparison with other models, we opted for the benchmark dataset,CaRB (Bhardwaj et al., 2019), used in previous studies, applying the same evaluation metric, F1 score."
>
> ---
>
> ### Question 2:
> The experimental setup is also weak.
>
> ### Answer 2:
> Although the space is limited and we have not shown too many experiments, we believe that our experimental setup is comprehensive and thorough. Specifically, our study incorporates four pivotal experiments:
>
> 1.	A direct comparison of our method with other SOTA techniques was carried out using the full training data. The results showcased that our proposed approach is comparable to the best-performing techniques under these settings.
> 2.	In the context of three low-resource settings, we benchmarked our approach against two types of techniques: labeling-based and generation-based. The evidence demonstrated that our method achieved SOTA performance in these low-resource conditions.
> 3.	A detailed contrast was drawn between our approach and the baseline, GEN2OIE. We meticulously reported and analyzed the performance metrics of every component of OK-IE. Empirical results indicate that, in comparison to the baseline, our approach has increased the F1 score from 31.2 to 52.9 and has notably expedited the convergence rate.
> 4.	We also tracked the performance trends of both generation-based methods and our proposed technique as training data scaled from limited quantities to the full dataset. This trend analysis vividly underscores that our proposed approach demands less data and converges at a swifter pace throughout the entire spectrum of training data.
>
> ---
>
> ### Question 3:
> The ablation experiment is also unclear for the reader.
>
> ### Answer 3:
> In the ablation study detailed in **Section 4.3** and depicted in **Table 5**, we transparently illustrate the impacts of OK-IE's individual components - task format transformation, anchor, and controlling order generation - on performance and convergence speed under low-resource settings. Moreover, we underscore this enhancement by investigating how the model converges to a result after just one single training epoch. We also provide a more detailed analysis of the ablation study in the **Appendix B.2** of our paper.
>
> ---
>
> ### Question 4:
>
> The tables could be just integrated in Figure 1.
>
> ### Answer 4:
>
> It's worth noting that in Figure 1, we track the performance of both generation-based methods and our proposed approach as training data increases from a small subset to the full amount. This curve provides an intuitive visualization of the changing trends.
>
> However:
>
> 1. Table 1 presents our method in contrast to other techniques under traditional settings. This table aims for an immediate comparison of the performance of various methods.
> 2. Tables 2 through 4 detail the performance of our proposed method against generation-based and labelling-based strategies, furnishing readers with a clear view of the performance and distinctive advantages of each category.
> 3. Table 5 showcases the incremental enhancements introduced by different components compared to the baseline. As part of the ablation study, it's imperative to distinctly lay out these nuances for a granular examination of our proposed methodology.
>
> In summary, given the disparate dependent variables across these tables, it's challenging, if not counterproductive, to integrate them into a single figure. Such an attempt could compromise clarity, obscuring the immediate insights that our current paper offers to readers.
>
> ---
>
> ### Question 5:
> The paper is too short and not well explained. I cannot understand the key component of the proposed idea, neither the working flow.
>
> ### Answer 5:
>
> Due to space constraints, our description might have lacked sufficient clarity. We will revise and refine the paper based on your suggestions in the revised version. To address your concerns immediately, we will provide a clearer explanation below.
>
>
> 1. ***The Problem***: In our paper, we address two main challenges in open information extraction. The first is the mismatch in task format between upstream and downstream tasks: While we can model the open information extraction task as a seq2seq problem, this is inconsistent with T5's pre-training task, namely span corruption. This inconsistency suggests that a significant amount of data is required to train our model to adapt to new task format. The second challenge is the 'order penalty'. Due to our reliance on teacher forcing during training, the model tends to rigidly perceive only one sequence as correct, unjustly penalizing other viable sequences. This requires more training time to converge to a stable state.
>
>     We've explained this in the beginning parts of our paper, specifically in the **Abstract** `L009-L013` and **Introduction** `L073-L084`. And if you're wondering why existed methods faced these issues, we've covered that too in the section **“Related work”** `from L154 to L172`.
>
> 2. ***Our Solution***: We've come up with two ways to handle these issues. The first involves a  transformation of the task format (as detailed in **section 3.1**). This approach effectively bridges the mismatch between the types of tasks in the upstream and downstream processes.  The second method （as detailed in **section 3.2**）introduces an 'Anchor' to explicitly specify which of the “subject, predicate, object" the model should generate at the sentinel position. This technique helps eliminate the 'order penalty'.
>
> 3. ***Proof***:  Not only did we propose these solutions, but we also validated them through comprehensive experiments. In the **Experiments** section, we show how our methods work well. We've also pinpointed exactly how our two techniques, the "task format transformation" and "controlling generation order", speed up training. We used less data and less time and still managed top-notch results, especially when resources were limited.
>
> 4. ***Working Flow Details***: If you're looking for a step-by-step guide for working flow, we've got you covered in **Appendix A.1**. And throughout our paper, we have provided real-world examples to give you a clearer picture of each part.

---

### Official Review · Reviewer_WhNg · 2023-08-11

**Soundness:** 3

**Excitement:**

3: Ambivalent: It has merits (e.g., it reports state-of-the-art results, the idea is nice), but there are key weaknesses (e.g., it describes incremental work), and it can significantly benefit from another round of revision. However, I won't object to accepting it if my co-reviewers champion it.

**Paper Topic And Main Contributions:**

This work introduces two improvements for an existing predicate extraction model GEN2OIE. The first technique is transforming the IE task into T5’s pre-training task span corruption to improve the data efficiency of the model. The second technique is the anchor method that instructs the model on the sequence order for generation. The experimental results show the effectiveness of the two proposed techniques.

**Questions For The Authors:**

- Which size of T5 is used in the experiments? Would the proposed methods have different improvement levels for models of different sizes?
- It would be helpful to report the zero-shot performance and compare OK-IE with removing the proposed methods. Especially given that span corruption is part of the pre-training task for T5, by simply transforming the task formulation, would the zero-shot performance show more difference compared to using traditional formulation?

**Reasons To Accept:**

- Significant empirical results show the effectiveness of the two proposed techniques, especially transforming task format to span corruption.
- The task format transformation technique is well motivated

**Reasons To Reject:**

- The authors claim that the proposed methods improve open IE, but the proposed techniques are based on a particular existing model (and keep the entire first stage of GEN2OIE), and only show its performance on the predicate extraction task. The claim does not match the evidence provided in the paper
- As the proposed methods are developed based on GEN2OIE, it would be nice to see the comparison with GEN2OIE directly for low-resource settings besides the comparison with IMOJIE

**Reproducibility:**

3: Could reproduce the results with some difficulty. The settings of parameters are underspecified or subjectively determined; the training/evaluation data are not widely available.

**Reviewer Confidence:**

3: Pretty sure, but there's a chance I missed something. Although I have a good feel for this area in general, I did not carefully check the paper's details, e.g., the math, experimental design, or novelty.

---

> ### Author Rebuttal · Authors · 2023-08-28
>
> **We sincerely appreciate the time and effort you dedicated to reviewing our work. Below, we respond to each of your comments and questions.**
>
> ---
>
> ### Question 1:
> The authors claim that the proposed methods improve open IE, but only show its performance on the predicate extraction task. The claim does not match the evidence provided in the paper.
>
> ### Answer 1:
>
> To begin with, it is imperative to clarify a critical misconception. In our study, what we presented is the performance of Open Information Extraction, not merely predicate extraction.
>
> Consistent with the methods we compared against and the datasets we employed; we conducted the open information extraction task, which extracts all triples, encompassing subject, predicate, and object, from given sentences. Notably, it's not confined to merely extracting the predicate. To illustrate, consider the example sentence provided in our paper `lines L196-L198`: “Elon Musk, who is the CEO of Tesla, also founded SpaceX”. Our proposed method can extract multiple triples such as '(Elon Musk; is the CEO of; Tesla)' and '(Elon Musk; founded; SpaceX)', rather than solely the predicates ‘is the CEO of’ and ‘founded’. In this instance, the common subject is 'Elon Musk', with paired predicates and objects being '(is the CEO of; Tesla)' and '(founded; SpaceX)', respectively. We've showcased the efficacy of our method in the open information extraction task within **Section 4 “Experiments”**.
>
>
> As mentioned in `lines L068-L072` of our paper:
> >"Considering the simplicity of predicate extraction from sentences, we retain the first stage of GEN2OIE and focus on optimizing the second stage, which is responsible for generating the complete triple ".
>
> ---
>
> ### Question 2:
> Would the proposed methods have different improvement levels for models of different sizes?
>
> ### Answer 2:
>
> Yes, the proposed methods exhibit varied improvement levels across different model sizes (base, large, xl, xxl). However, the differences in these improvements are minimal.
>
> Nevertheless, it's necessary to underline that the core objective of this paper centers on enhancing model training efficiency within low-resource settings. Taking into account that increasing the model's parameter scale leads to a marked surge in memory consumption, ranging from 990MB to 45GB, this deviates from our fundamental goals of **"low resource"** and **"efficiency"**. Due to this rationale, such results were not showcased in the main body of the paper. We intend to incorporate this data in the appendix of the revised version.
>
> ---
>
> ### Question 3:
> Especially given that span corruption is part of the pre-training task for T5, by simply transforming the task formulation, would the zero-shot performance show more difference compared to using traditional formulation?
>
> ### Answer 3:
>
> Yes, in the zero-shot setting, our method that incorporates the task format transformation demonstrates a notable performance difference when compared to the traditional formulation.
>
> In our previous experiments, specifically with T5-xl and T5-xxl models, we found that without this transformation, the models had difficulty in generating well-structured triples. However, when applying our proposed **'task format transformation'**, they successfully generated the desired triples. In contrast, models with fewer parameters did not show significant improvement, whether the transformation was applied or not.
>
> Considering the main focus of our paper is on enhancing model **training** efficiency in low-resource settings, we did not include these zero-shot experiment results in the main content. However, we plan to provide this data in the appendix of our revised version.
>
> ---
>
> ### Question 4:
> As the proposed methods are developed based on GEN2OIE, it would be nice to see the comparison with GEN2OIE directly for low-resource settings besides the comparison with IMOJIE.
>
> ### Answer 4:
> Yes, we compare with GEN2OIE directly in low-resource setting. In **Section 4.3, "Ablation Studies on System Components,"** and as detailed in Table 5, we provide an in-depth comparison between OK-IE and GEN2OIE in low-resource settings.
> When compared to GEN2OIE, OK-IE boosted the F1 score in low-resource settings from 31.2 to 52.9 and accelerated the rate of convergence.
>
> ---
>
> ### Question 5:
> It would be helpful to report the zero-shot performance and compare OK-IE with removing the proposed methods.
>
> ### Answer 5:
> Thanks for your suggestion. Based on Answer 3, we plan to supplement our appendix with additional experimental data showcasing zero-shot performance across the four parameter sizes—base, large, xl, and xxl. Specifically, this will detail the impact of utilizing versus not utilizing the “task format transformation” on their performance. We will provide detailed description in the revised version.
>
> ---
>
> ### Question 6:
> The proposed techniques are based on a particular existing model.
>
> ### Answer 6:
>
> In the OpenIE task, leveraging an existing pretrained model has proven to enhance performance and reduce the requisite training data. Therefore, both OK-IE (ours) and GEN2OIE (the baseline for this paper) are based on the T5 model.
>
> ---
>
> ### Question 7:
> Which size of T5 is used in the experiments?
>
> ### Answer 7:
> We utilized the T5 "base" configuration, comprising 220 million parameters. This is elaborated in Appendix B.1 "Experimental Setup Details".
>
> ---
>
> ### Question 8:
> The proposed techniques keep the entire first stage of GEN2OIE.
>
> ### Answer 8:
>
> The crux of Open Information Extraction lies in extracting the full, cohesive triple, while predicate extraction is more straightforward.
>
> As stated in `lines L068-L072` of our paper: "Considering the simplicity of predicate extraction from sentences, we retain the first stage of GEN2OIE and focus on optimizing the second stage, which is responsible for generating the complete triple."
>
> Moreover, in **Appendix A2** of our paper, we have meticulously delineated the distinctions between OK-IE and GEN2OIE. It becomes evident that in aspects such as data requirements for training, generation sequence, generation efficiency, and format, OK-IE manifests clear advantages and differentiation.

---

### Meta-Review · Area_Chair_JPJA · 2023-09-19

**Recommendation:** 2

**Metareview:**

This research paper explores open information extraction and introduces an approach aimed at addressing the problem of data inefficiency. The study proposes two enhancements for the GEN2OIE predicate extraction model. The first enhancement involves reconfiguring the information extraction task to align with T5's pre-training task known as "span corruption," which significantly enhances the model's data efficiency. The second improvement is the introduction of an anchor method that guides the model in determining the sequence order for generation. In summary, this paper builds upon a previous method and achieves significant advancements within a particular module.

---

### Decision · Program_Chairs · 2023-10-07

**Decision:**

Accept-Findings

**Comment:**

This research paper explores open information extraction and introduces an approach aimed at addressing the problem of data inefficiency. The study proposes two enhancements for the GEN2OIE predicate extraction model. The first enhancement involves reconfiguring the information extraction task to align with T5's pre-training task known as "span corruption," which significantly enhances the model's data efficiency. The second improvement is the introduction of an anchor method that guides the model in determining the sequence order for generation. In summary, this paper builds upon a previous method and achieves significant advancements within a particular module.